# Two-Stage Grid-Connected Frequency Regulation Control Strategy Based on Photovoltaic Power Prediction

**Shuzheng Wang \*, Haiming Zhu and Shaowen Zhang**

School of Electrical Engineering, Nanjing Institute of Technology, Nanjing 211167, China
\* Correspondence: shuzheng_wang@njit.edu.cn

**Abstract:** The large number of photovoltaics connected to the distribution network via power electronic converters squeezes the functional space of traditional synchronous generators in the power system and reduces the inertia of the network itself. However, due to the random and fluctuating nature of PV power generation, different types of meteorological conditions can also affect the inertia support capability of PV output power. Therefore, this paper proposes a frequency regulation control strategy based on the dynamic characteristics of the grid-side DC capacitor. Firstly, the control strategy of the grid-side inverter is improved and the mechanism of the frequency dynamic response model under PV penetration is analysed. Secondly, data from different weather types are correlated and analysed to predict PV power, and wireless sensor technology is used to introduce the data signals into the control link. Finally, a simulation model of a two-stage PV power generation system is developed and the fast-response capability of the system under different control parameters when the load is increased or decreased is analysed. The results show that under the proposed frequency regulation strategy, the larger the regulation coefficient and virtual inertia time constant, the greater the virtual inertia provided by the PV power generation, which improves the stability of the distribution network.

**Keywords:** two-stage photovoltaic power generation systems; wireless sensing; power prediction; grid-side capacitance dynamic characteristics; frequency dynamic characteristics; frequency regulation

## 1. Introduction

In the context of the global energy transition, renewable energy sources such as photovoltaic power systems are connected to the distribution network to participate in its operation [1–3]. With the emergence of a high proportion of power electronic devices such as PV grid-connected inverters, the number of conventional generating units has been drastically reduced and the resulting multiple rotational inertia no longer exists, prompting an increase in grid frequency shifts and even collapses [4,5]. There is an urgent need for PV power systems containing devices such as power electronic converters to actively participate in frequency regulation. At the same time, it is more beneficial to regulate the frequency of the system by analysing different meteorological information and establishing a mapping model between meteorological data and PV power so as to accurately predict the PV output power.

At present, there are two main types of frequency regulation methods for photovoltaic power generation. One is to operate at the maximum power point, and release or absorb active power through energy storage equipment, so as to provide support inertia for the system to participate in frequency regulation. However, the photovoltaic system does not operate through load shedding or reserve power at this time and lacks the active power required to adjust the frequency. The frequency regulation ability is limited and the system frequency stability state after disturbance cannot be maintained or restored. The second type is the non-energy storage operation mode of the photovoltaic power generation system, that is, the photovoltaic array operates at the non-maximum power point, and the

influence of the system active power fluctuation on the frequency is reduced by reserving the reserve power. Both frequency modulation methods can adopt droop control or virtual synchronous generator control and other frequency modulation strategies to smooth the system frequency fluctuation.

In terms of droop control or load shedding control operation, [6] proposed a control scheme which can realise seamless switching between the grid-connected mode and the island mode of microgrid under the condition of system frequency fluctuation, and can reduce the fluctuation of system frequency in the common coupling point of the microgrid and public grid, which is helpful in promoting the research and development of new control schemes in the field of the microgrid. The authors of [7] proposed an adaptive gain control strategy, which established a functional relationship between the gain and the system frequency. It can release more support inertia through primary frequency modulation and can last longer and restore the optimal operating point of the system frequency during the secondary frequency modulation. In [8], a tracking control strategy based on a quadratic linear regulator was proposed, which was different from the traditional droop control. It can adjust the frequency of the system in real time under the condition of given inertia and droop coefficient and is effectively verified by the full signal photovoltaic power generation model. In [9], a system frequency control strategy based on an event-triggered mechanism was proposed. This strategy can give the constraint condition of nominal frequency, so that the microgrid can check the trigger condition according to its own clock cycle, which improves the efficiency of microgrid frequency regulation. In [10], the dynamic characteristics of the DC/DC converter, DC/AC converter, AC side filter, and virtual inertial controller were considered by establishing a small signal analysis model. The sensitivity of design parameters to the stability of the grid-connected virtual inertial photovoltaic system was analysed by eigenvalues. The authors of [11] established a mathematical model of system frequency deviation and reserved power of photovoltaic power generation system and designed a proportional load shedding controller. When photovoltaic systems with different reserve capacities are connected to the grid, the load shedding controller will adjust the system frequency at different levels according to the reserve power ratio, so as to maintain the stable operation of the system. In [12], the traditional droop control was also improved, and the frequency deviation and frequency change rate were suppressed by using the fast power compensation strategy to adjust the frequency of the system. The authors of [13,14] proposed a frequency modulation control strategy based on load shedding technology. This strategy combined the maximum power tracking algorithm, tracked the power in real time through curve fitting, and predicted the light intensity to implement better frequency regulation. The literature in [15] is based on distributed energy storage. By establishing the mathematical function model of the distributed energy storage and photovoltaic power generation system, the frequency regulation strategy of distributed energy storage participating in the photovoltaic power generation system is proposed. The strategy uses the power frequency characteristic curve of the system for analysis, which has the effect of smoothing frequency fluctuation and reducing frequency deviation. It has reference significance for subsequent energy storage participating in photovoltaic system frequency regulation research.

In the aspect of virtual synchronous generator frequency modulation control, based on the virtual synchronous generator technology, the photovoltaic power generation and energy storage grid-connected system was established in the literature [16], and the autonomous PI fuzzy control algorithm applied to the double closed-loop control was proposed. The strategy can independently adjust the DC voltage and maintain the stability of the system frequency. In [17], a control strategy of an interconnected converter based on virtual synchronous generator technology was proposed. This control strategy can make the hybrid microgrid seamlessly switch between grid-connected operation mode and island operation mode, and automatically adjust the operation mode, so that the system frequency and DC voltage remain at the rated value, that is, to achieve no difference adjustment to maintain the safe and stable operation of the system. The authors of [18] proposed an

inertia damping adaptive virtual synchronous generator control strategy. This strategy established a functional relationship between the frequency parameters and the rotational inertia of the virtual synchronous generator, and selected the optimal damping coefficient value to achieve rapid response of the system frequency. The system frequency change rate and frequency deviation are reduced, and the system dynamic adjustment capability is optimised. In [19], an operation and control strategy based on centralised photovoltaic power generation was proposed that can provide inertial response and primary frequency control to support the black start of large capacity power systems. The proposed strategy can only provide inertial response and primary frequency control support during system recovery, which improves the dynamic- and steady-state performance of the frequency response of the photovoltaic system. The authors of [20,21] extended from a single virtual synchronous generator to multiple parallel operations. Taking the parallel virtual synchronous generator as the research object, considering the interaction and influence of its parallel operation, the linear impedance and control parameters of the system power and load power are optimised, and the frequency deviation of the parallel system is effectively controlled. Based on the microgrid system formed by photovoltaic energy storage, the improved control strategies related to communication and energy storage grid connection were summarised in the literature [22]. It can realise seamless switching and island operation, which is more conducive to the stability of important parameters such as system frequency. In [23], the two-stage photovoltaic power generation system was taken as the research object, and the fault ride-through control strategy was proposed. By limiting the active power of photovoltaic output, the improvement of the main power-quality indicators such as frequency and voltage of the system is ensured.

In this paper, a new control method for frequency regulation is proposed in order to introduce the inertia and frequency regulation capability of a two-stage PV power generation system connected to a distribution network without energy storage. (1) The active frequency regulation capability of the distribution system is achieved by using the derived DC voltage and frequency relationship to release the reserved PV power. This is a simple method that does not require the use of irradiance measurements. (2) Unlike traditional methods of regulating frequency through support power only, the proposed method allows for more accurate prediction of the output active power and better cost savings. (3) Timely wireless sensor signals can prevent DC voltage collapse when the active power required is greater than the backup power of the PV system.

## 2. System Topology and General Control Strategy

### 2.1. Photovoltaic Array Structure

As depicted in Figure 1, the PV array consists of $N_p$ columns with $N_s$ PV panels per column, and $V_{PV}$ and $I_{PV}$ are the DC voltage and current for the PV array, respectively.

The equations of the PV array output current $I_{PV}$ and output power $P_{PV}$ versus output voltage $V_{PV}$ are as follows [24]:

$$\begin{cases} I_{PV} = N_p I_{sc} \left[ 1 - C_1 \left( e^{\frac{V_{PV}}{C_2 N_s U_{OC}}} - 1 \right) \right] \\ P_{PV} = N_p I_{sc} \left[ 1 - C_1 \left( e^{\frac{V_{PV}}{C_2 N_s U_{OC}}} - 1 \right) \right] \cdot V_{PV} \end{cases} \tag{1}$$

where $I_{sc}$ and $U_{OC}$ are the short-circuit current and open-circuit voltage for the PV panel, and $C_1$ and $C_2$ are coefficients to be determined.

### 2.2. Two-Stage Photovoltaic Power System Structure

As displayed in Figure 2, the two-stage PV system structure contains a PV array, Boost converter, PV inverter, line load, transformer, and AC grid [25].

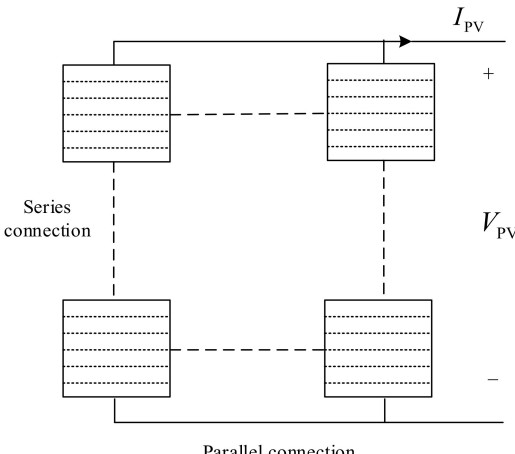

**Figure 1.** PV array structure.

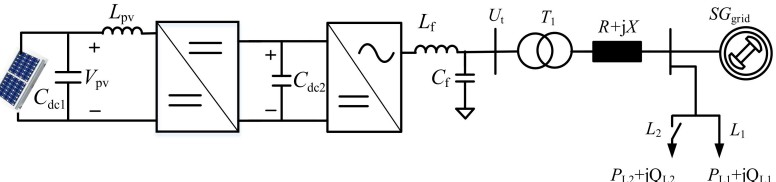

**Figure 2.** Topology for two-stage PV system.

In Figure 2, $C_{dc1}$ and $C_{dc2}$ are the filter capacitors, and $U_t$ and $U_g$ are the parallel network voltage and grid voltage, respectively. Compared with the single-stage PV system, the two-stage system adds a Boost converter on top of it, reducing the PV output power's volatility and randomness, stabilising the high-voltage-side DC voltage, and, finally, connecting the AC output to the grid through the grid-side inverter.

### 2.3. Conventional Control for Boost Converter

The Boost converter reduces electromagnetic interference to the PV array, while widening the MPPT range and increasing the efficiency of PV generation.

Figure 3 presents the control strategy of the Boost converter. The output voltage and current of the PV array are output by the MPPT with the DC voltage reference $V_{PVref}$, and the error signal is output by the PI controller with the duty cycle $d$, which drives the normal operation of the Boost converter.

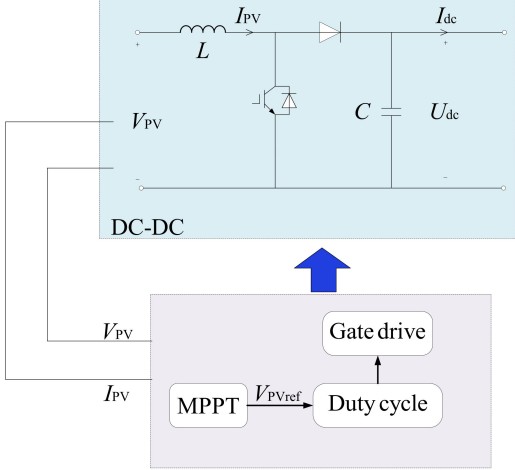

**Figure 3.** Conventional control for Boost converter (The blue arrow represents the driver signal in the control link that is transmitted to the IGBT gate).

## 3. Dynamic Characteristics of Grid Frequencies under PV Penetration

The dynamic characteristics of the grid frequency refer to the rise or fall of the system frequency when unbalanced power occurs in the system, and the operational security for the grid will be at risk. The grid frequency model is a closed-loop control system that includes the governor, prime mover, synchronous generator, and load. Conventional synchronous generators, whether thermal or hydro, are represented by the closed-loop system in Figure 4. In Figure 4, $R$ is the regulation factor, $T$ is the equivalent inertia time constant of the prime mover, $F_{HP}$ is the proportion of work performed by the high-pressure cylinder of the prime mover, $H$ is the inertia time constant of the synchronous generator set, and $D$ is the damping factor.

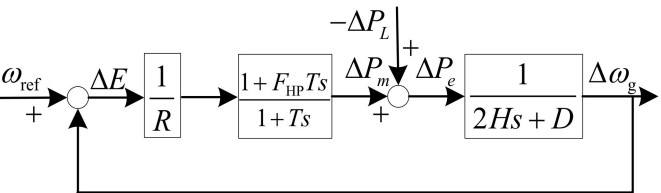

**Figure 4.** Conventional response model for grid frequency.

When PV systems are connected to the grid, the conventional grid frequency model needs to introduce a parameter characterising the share of conventional synchronous generating units or PV systems, as given in Equation (2):

$$r = \frac{\text{Power for conventional generators (MW)}}{\text{Power for system loads (MW)}} \tag{2}$$

where $r$ denotes the proportion coefficient of conventional synchronous generating units, with the PV system penetration rate $1 - r$, where $0 \leq r \leq 1$.

The model in Figure 4 is based on the capacity of the generator (or turbine). If instead the system load power is used as the reference value, then both the rotor time constant of the synchronous unit and the output power of the turbine should be multiplied by a factor $r$. After introducing the factor $r$ into the conventional grid closed loop system, the grid frequency model under PV penetration is shown in Figure 5.

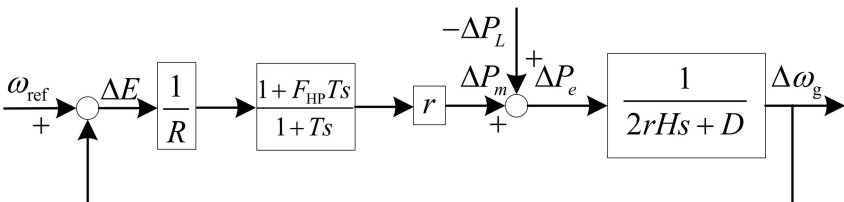

**Figure 5.** Grid frequency response model for photovoltaic system penetration.

The transfer function $G(s)$ from the load perturbation $\Delta P_L$ to the frequency perturbation $\Delta f_g$ is:

$$\begin{aligned} G(s) &= \frac{-R(1+Ts)}{R \cdot (1+Ts) \cdot (2rHs+D) + r \cdot (1+FTs)} \\ &= G_0 \frac{s+z_0}{s^2 + 2\zeta\omega_n s + \omega_n^2} \end{aligned} \tag{3}$$

where $\zeta$ is the damping ratio, $\omega_n$ is the undamped oscillation frequency, $G_0$ is the scaling factor, and $z_0$ is the inverse of the reheat time constant of the prime mover.

When a step disturbance of load occurs in the system, then the grid frequency in the frequency domain is:

$$
\begin{aligned}
f_{\text{r\_pu}}(s) &= f_{\text{ref\_pu}}(s) + \frac{G(s)}{s} \\
&= \frac{1}{s} + G_0 \left\{ \frac{1}{\omega_d} \cdot \frac{\omega_d}{(s+\zeta\omega_n)^2+\omega_d^2} + \frac{z_0}{\omega_n^2} \cdot \right. \\
&\qquad \left. \left[ \frac{1}{s} - \frac{s+\zeta\omega_n}{(s+\zeta\omega_n)^2+\omega_d^2} - \frac{\zeta\omega_n}{\omega_d} \frac{\omega_d}{(s+\zeta\omega_n)^2+\omega_d^2} \right] \right\}
\end{aligned}
\tag{4}
$$

where $f_{\text{r\_pu}}(s)$ is the grid frequency reference, $f_{\text{ref\_pu}}(s)$ is $1/s$, $G(s)/s$ is the frequency disturbance value and $\omega_d$ is the natural damping frequency. The grid frequency in the time domain is obtained after the pull-type inverse transformation of Equation (4) as:

$$
f(t) = 1 - \frac{1}{2rH} \left[ \frac{z_1}{\omega_n^2} - e^{-\zeta\omega_n t} A \sin(\omega_d \cdot t + \beta) \right]
\tag{5}
$$

$$
\begin{cases}
\omega_d = \omega_n \cdot \sqrt{1-\zeta^2} \\
z_1 = 1/T \\
A = \sqrt{\left(\frac{z_1}{\omega_n^2}\right)^2 + \left(\frac{-z_1 \cdot \zeta + \omega_n}{\omega_n \cdot \omega_d}\right)^2} \\
\beta = \arctan\left[ \frac{\omega_d \cdot z_1}{(z_1 \cdot \zeta - \omega_n) \cdot \omega_n} \right]
\end{cases}
\tag{6}
$$

where $\omega_d$ is the damped oscillation frequency.

On account of the increasing penetration of photovoltaic systems, the stationary photovoltaic components lead to a reduction in the equivalent inertia for the system, and the frequency characteristics for the grid will gradually deteriorate, increasing the frequency variation rate of the receiving grid, increasing the frequency steady-state deviation, reducing the frequency nadir, etc., which seriously endangers the safe operation of the power system.

## 4. Analysis of Frequency Regulation Control Strategy for PV and Its Mechanism

### 4.1. FR Control to the Left for the MPP

As shown in Figure 6, M and N are the power points on the left and right sides, respectively, of the PV array operation when the PV system leaves a certain amount of backup power. When there is an imbalance between the power output of the synchronous generator set and the load power, the system frequency will fluctuate significantly. Under the action of the unbalanced power, the equation of rotor motion in the synchronous generator set is illustrated in Equation (7) [26]:

$$
\begin{cases}
\frac{d\delta}{dt} = (\omega - 1)\omega_0 \\
2H_{\text{SG}}\omega \frac{d\omega}{dt} = 2H_{\text{SG}} f \frac{df}{dt} = P_m - P_e
\end{cases}
\tag{7}
$$

where $\delta$ is the rotor work angle, $\omega_0$ is the rated electrical rotor angular velocity, $\omega$ is the rotor electrical angular velocity, and $H_{\text{SG}}$ is the rotor inertia time constant.

From Figure 2, the dynamic equation for the DC capacitor of the grid-connected side of the PV inverter is:

$$
P_{\text{dc}} - P_g = C_{\text{dc2}} \frac{dU_{\text{dc}}}{dt} \cdot U_{\text{dc}}
\tag{8}
$$

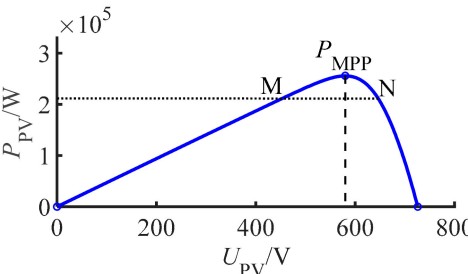

**Figure 6.** PV characteristic curve for PV array.

The inertial response of a synchronous generator set can therefore be simulated dynamically with the grid-side DC capacitor, which gives:

$$C_{dc2}U_{dc}\frac{dU_{dc}}{dt} = 2H_{dc}f\frac{df}{dt} \tag{9}$$

$$\int_{U_{dc0}}^{U_{dc}} C_{dc2}U_{dc}\frac{dU_{dc}}{dt} = \int_{f_0}^{f} 2H_{dc}f\frac{df}{dt} \tag{10}$$

$$\frac{1}{2}C_{dc2}\left(U_{dc}^2 - U_{dc0}^2\right) = H_{dc}\left(f^2 - f_0^2\right) \tag{11}$$

where $H_{dc}$ is the virtual inertia time constant, $U_{dc0}$ is the initial value of the grid-side DC capacitor voltage, and $f_0$ is the nominal system frequency of 50 Hz. Linearisation of Equation (11) yields:

$$\begin{cases} C_{dc2}U_{dc0}\Delta U_{dc} = 2H_{dc}f_0\Delta f \\ \Delta U_{dc} = U_{dc} - U_{dc0} \\ \Delta f = f - f_0 \end{cases} \tag{12}$$

$$\begin{cases} \Delta U_{dc} = k\Delta f \\ k = \frac{2H_{dc}f_0}{C_{dc2}U_{dc0}} \end{cases} \tag{13}$$

From Equation (13), the regulation factor $k$ and the virtual inertia time constant $H_{dc}$ are important influencing parameters of the control strategy proposed in this paper, and it can be seen that the regulation factor $k$ and the virtual inertia time constant $H_{dc}$ are proportional to the incremental DC voltage $\Delta U_{dc}$ on the grid-connected side of the inverter. When the regulation factor $k$ and the virtual inertia time constant $H_{dc}$ increase, $\Delta U_{dc}$ then increases and the more support power the PV plant can provide to the system, which can weaken the volatility in the dynamic process of the system frequency. Therefore, we can improve the grid-side inverter control strategy, as shown in Figure 7.

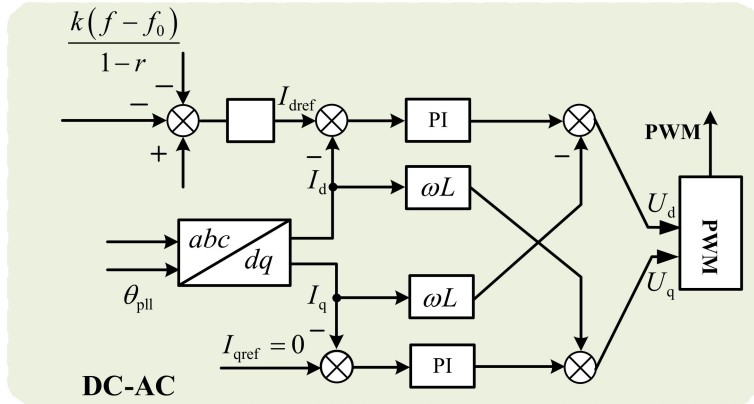

**Figure 7.** Improved control strategy of outer loop of inverter.

Since the literature [27] focuses on the improvement of the control strategy and lacks an analysis of the deeply influential parameters in terms of control, this paper, based on the FR control strategy proposed in the literature [27], firstly further improves the control strategy by introducing the PV permeability parameter $1 - r$, which can be adaptively adjusted for the numerical size of the grid-connected PV capacity, and then analyses the improved control through the frequency response model and the Lyapunov energy function. The improved control is then mechanistically analysed by means of a frequency response model and a Lyapunov energy function, and the influence of different important parameters on the stability of the system is investigated.

From the above analysis, the amount of change in voltage on the DC side affects the amount of active power absorbed by the shunt capacitors of the grid-connected converter, which in turn has an impact on the active power of the whole system. By adding virtual inertia control to the grid-connected converter, the inertia of the system is increased by changing the parameter $k$ in the control law. The system increases its inertia by changing the parameter $k$ in the control law, which in turn increases the inertia when frequency fluctuations occur. In addition to the frequency support from the synchronous generator, this part can also support the frequency when frequency fluctuations occur. This reduces the frequency variation bias by changing the parameter $k$ in the control law to increase the inertia of the system. This reduces the frequency deviation.

As the grid-side capacitor voltage and the PV array-side capacitor voltage correlate, i.e., $V_{PV} = k_f U_{dc}$, the corrected grid-side DC voltage will be transmitted to the low-voltage-side DC voltage, and the fluctuation of the low-voltage-side voltage will increase the output power of the PV array. The control of each converter thus achieves the purpose of frequency regulation.

When the PV array output power runs to the left of the maximum power point M, as described in Figure 6, the output power can be treated using linear fitting means, as described in Equation (14):

$$P = aV_{PV} + b \tag{14}$$

where both $a$ and $b$ are fitting coefficients, then, linearisation of Equation (13) gives:

$$\begin{cases} \Delta P = a\Delta V_{PV} = a \cdot k_f \Delta U_{dc} = a \cdot k_f k \Delta f \\ G(s) = \frac{\Delta P}{\Delta f} = a \cdot k_f k \end{cases} \tag{15}$$

From Equation (15), the transfer function link $G(s)$ is added to the conventional grid frequency response model, so the grid frequency response model under PV penetration is pictured in Figure 8.

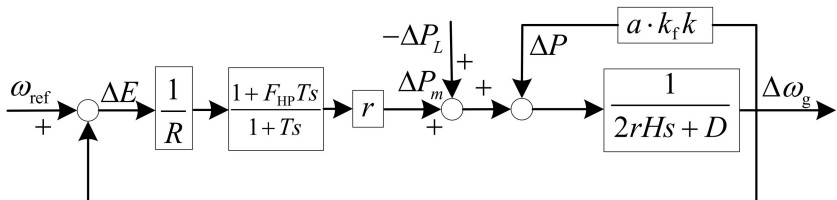

**Figure 8.** Grid frequency response model to the left of the MPP.

From Figure 8, the transfer function $G_1(s)$ from the load perturbation $\Delta P_L$ to the frequency perturbation $\Delta f_g$ is as follows:

$$G_1(s) = \frac{-R(1 + Ts)}{R \cdot (1 + Ts) \cdot [2rHs + (D + ak_f k)] + r(F_{HP}Ts + 1)} \tag{16}$$

Therefore, compared to the conventional grid frequency response model, the essence of this control strategy of frequency regulation is to change the damping factor $D$ of the synchronous generator set, with the new damping factor becoming $D + ak_f k$.

According to Equation (15), under different control coefficients $k$, the relationship between the initial value $df/dt$ of grid frequency change rate and the minimum drop value $f_{\text{peak}}$ of grid frequency and the inertial time constant $H_{\text{dc}}$ is shown in Figure 9. From Figure 9, it can be seen that the change of control coefficient $k$ has no effect on the initial value of grid frequency change rate; as the time constant $H_{\text{dc}}$ increases, the lowest frequency drop increases.

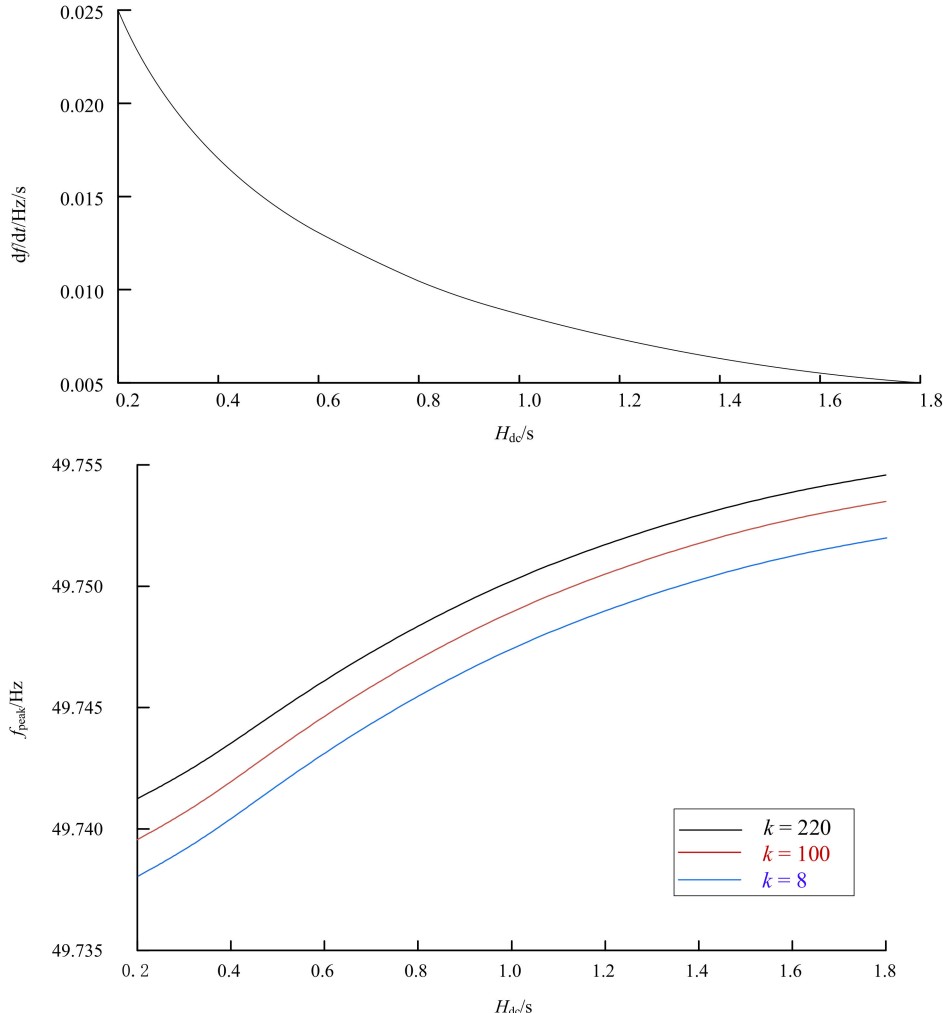

**Figure 9.** Relation curve between frequency characteristic quantity and $H_{\text{dc}}$.

### 4.2. FR Control to the Right for the MPP

The two-stage PV system will operate to the left and right of the maximum power point, and it can be set up as depicted at point N in Figure 6 when the PV array is operating to the right of the maximum power point. As can be seen from the *P-U* curve in Figure 6, a linear fit cannot be made to the right half at this point.

Using the dynamics of the grid-side DC capacitor to simulate the inertial response of a synchronous generator set, this process can be linearised, i.e., by collapsing Equation (8) to give:

$$\begin{aligned} \Delta P &= C_{\text{dc2}} U_{\text{dc0}} \cdot s \Delta U_{\text{dc}} \\ &= C_{\text{dc2}} U_{\text{dc0}} \cdot s k \Delta f \end{aligned} \tag{17}$$

From Section 2.1, it can be seen that the DC voltage on the PV array side is maintained at a constant value, and, thus, the PV array output power is maintained at a steady-state value. This process can be seen as a result of virtual inertia control and therefore yields:

$$\begin{aligned} \Delta P &= 2H_{\mathrm{PV}}\frac{\mathrm{d}f}{\mathrm{d}t} \\ &= 2H_{\mathrm{PV}}{\cdot}s\Delta f \end{aligned} \tag{18}$$

From Equations (16) and (17), it can be seen that the increased virtual inertia corresponds to the superposition of the two processes described above, thus adding a transfer function link to the traditional grid frequency response model:

$$G(s) = \frac{\Delta P}{\Delta f} = (k{\cdot}C_{\mathrm{dc2}}U_{\mathrm{dc0}} + 2H_{\mathrm{PV}}){\cdot}s \tag{19}$$

From Figure 10, the corresponding transfer functions are:

$$G_2(s) = \frac{-R(1+Ts)}{R{\cdot}(1+Ts){\cdot}[r(2H + kC_{\mathrm{dc2}}U_{\mathrm{dc0}} + 2H_{\mathrm{PV}})s + D] + r(F_{\mathrm{HP}}Ts + 1)} \tag{20}$$

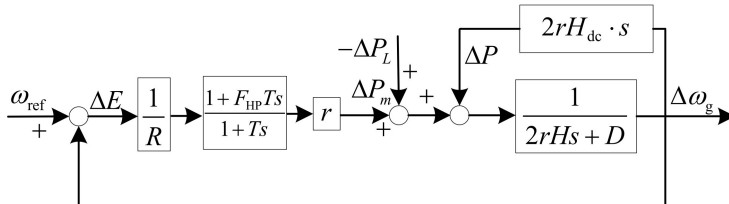

**Figure 10.** Grid frequency response model to the right of the MPP.

The essence of this control strategy is, therefore, to alter the inertia time constant $H$ for the synchronous generator set, and the novel virtual inertia time constant transforms $H_{\mathrm{dc}} = 2H + kC_{\mathrm{dc2}}U_{\mathrm{dc0}} + 2H_{\mathrm{PV}}$.

Models of power systems:

$$\begin{cases} 2H\dfrac{\mathrm{d}^2\Delta\delta_{\mathrm{g}}}{\mathrm{d}t^2} = \Delta P_{\mathrm{m}} - \underbrace{\dfrac{U_{\mathrm{g}}U_{\mathrm{t}}}{X_{\mathrm{g}}}\sin\Delta\delta_{\mathrm{g}}}_{\Delta P_{\mathrm{L}}} - D\Delta\dot\delta_{\mathrm{g}} \\[4mm] \Delta P_m = \left(P_0 - \dfrac{\Delta\dot\delta_{\mathrm{g}}}{R}\right)\dfrac{1 + F_{\mathrm{HP}}Ts}{1+Ts} \end{cases} \tag{21}$$

$\Delta P_{\mathrm{m}}$ comes from the generator and the virtual inertial control of the PV can affect the system equivalents $H$ and $D$. Further, the Lyapunov energy function can be constructed as:

$$V\left(\delta_{\mathrm{g}}, \dot\delta_{\mathrm{g}}\right) = E_{\mathrm{k}} + E_{\mathrm{p}} = H\dot\delta_{\mathrm{g}}^2 + \left(E_0 - P_0\delta_{\mathrm{g}} - \frac{U_{\mathrm{g}}U_{\mathrm{t}}}{X_{\mathrm{g}}}\cos\delta_{\mathrm{g}}\right) \tag{22}$$

where $E_{\mathrm{k}}$ and $E_{\mathrm{p}}$ are the kinetic and potential energies, respectively, of the system. Derivation of the Lyapunov energy function yields:

$$\begin{cases} \dot V\left(\delta_{\mathrm{g}}, \dot\delta_{\mathrm{g}}\right) = 2H\dot\delta_{\mathrm{g}}\ddot\delta_{\mathrm{g}} - P_0\dot\delta_{\mathrm{g}} + \frac{U_{\mathrm{g}}U_{\mathrm{t}}}{X_{\mathrm{g}}}\sin\delta_{\mathrm{g}}\dot\delta_{\mathrm{g}} = -\left(D + \frac{1}{R}\right)\dot\delta_{\mathrm{g}}^2 - \dot\delta_{\mathrm{g}}\beta \\[3mm] \beta = \frac{1}{R}\frac{(F_{\mathrm{HP}}-1)Ts}{1+Ts}\dot\delta_{\mathrm{g}} \end{cases} \tag{23}$$

where $F_{\mathrm{HP}} \in (0, 1)$, so that we have:

$$\dot V\left(\delta_{\mathrm{g}}, \dot\delta_{\mathrm{g}}\right) = -\left(D + \frac{1}{R}\right)\dot\delta_{\mathrm{g}}^2 - \dot\delta_{\mathrm{g}}\beta \le -D\dot\delta_{\mathrm{g}}^2 < 0 \tag{24}$$

That is, Equation (23) is negative definite. According to the principle of Lasalle invariance, there is local stability in the system. Therefore, the domain of attraction of the system can be estimated by taking the energy at the point of unstable equilibrium as the boundary value:

$$\Omega_{\mathrm{L}} = \left\{ V\left(\delta_{\mathrm{g}}, \dot{\delta}_{\mathrm{g}}\right) < L, L > 0, \dot{V}\left(\delta_{\mathrm{g}}, \dot{\delta}_{\mathrm{g}}\right) < 0 \right\} \tag{25}$$

Firstly, Figure 11 shows the effect of the equivalent inertia $H$ on the system's domain of attraction (i.e., the domain of stability). It is noticed that an increase in inertia reduces the area of the system's domain of attraction, meaning that it is detrimental to the stability of the system. In other words, although inertia reduces the rate of change of the frequency and the equilibrium point translation range of the system when subjected to external perturbations, the system frequency takes longer to adjust to the steady-state value. On the other hand, a small inertial system has a larger area of the attraction domain, but the system can be shifted significantly when disturbed, which can also cause the system to escape from the attraction domain and cause system instability. The control rate of virtual inertia therefore requires a trade-off between the attraction domain and the immunity of the system to find a compromise.

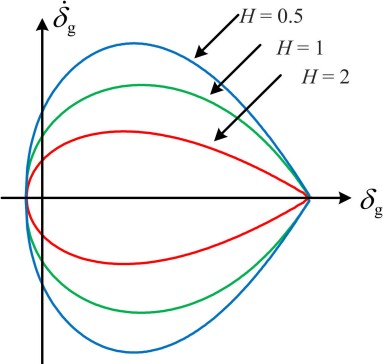

**Figure 11.** The effect of $H$ on the attraction domain.

Figure 12 illustrates the effect of the equivalent damping factor $D$ on the attraction domain of the system. An increase in damping increases the area of the attraction domain of the system and improves the stability of the system. In addition, the damping factor can increase the equivalent damping ratio of the system, which can suppress the power and frequency oscillations of the synchronous generator. Therefore, by providing primary frequency regulation for the system through PV load shedding operation, the stability and dynamic performance of the system can be effectively improved and the oscillations and frequency dips of the system can be suppressed.

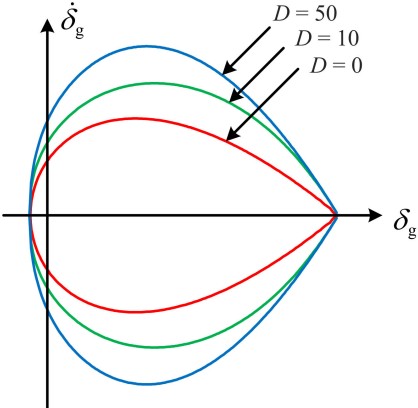

**Figure 12.** The effect of $D$ on the attraction domain.

At the same time, there is hysteresis in the governor of the generator, i.e., $F_{HP} < 1$. In the ideal case, if $F_{HP} = 1$ and there is no hysteresis in the governor, the equivalent damping of the power system is $D + 1/R$, which can further increase the stability of the system.

## 5. PV Power Prediction Algorithm

### 5.1. Selection of Datasets and Correlation Coefficients

In this paper, PV historical power generation data are used, and the input data $x_i^t$ represent the *i*-th variable of the time *t* data. In order to avoid the adverse effects of different dimensions, this paper uses Equation (25) to normalise each variable to [0, 1], namely:

$$x' = \frac{x - x_{min}}{x_{max} - x_{min}} \tag{26}$$

Feature selection can effectively select input variables closely related to photovoltaic power, avoid interference of unrelated variables on model power prediction, and reduce computational complexity. The Pearson correlation coefficient is used to measure the correlation between each variable and photovoltaic power.

The formula of the Pearson correlation coefficient is:

$$\rho_{x,y} = \frac{\sum\limits_{i=1}^{n} (x_i - \overline{x})(y_i - \overline{y})}{\sqrt{\sum\limits_{i=1}^{n} (x_i - \overline{x})^2} \sqrt{\sum\limits_{i=1}^{n} (y_i - \overline{y})^2}} \tag{27}$$

where the Pearson correlation coefficient values between solar irradiance and PV module temperature and PV power are $\rho_1$ and $\rho_2$, respectively. It has strong correlation.

### 5.2. SOM Clustering Algorithm

The SOM clustering algorithm is a single-layer neural network based on competitive learning, and using the SOM self-organising clustering method, it is able to classify PV data into three types of weather data: sunny days, rainy days, and sudden-change days. By improving the impact of the network on PV power fluctuation characteristics, through this SOM network calculation, it can find the winning node:

$$d_j = \|X - W_{ij}\| = \sqrt{\sum\limits_{i=1}^{n} (x_i - W_{ij}(t))^2} \tag{28}$$

$$\|X - W_c\| = \min\{d_j\}$$

where $W_{ij}$—weight between node *i* and node *j*; $W_c$—weight of the winning node.

Parameter adjustment $W_{ij}$:

$$W_{ij}(t+1) = W_{ij}(t) + \eta(t)h_{c,j}(t)(X_i - W_{ij}(t)) \tag{29}$$

where $\eta(t)$—learning rate, a gain function; $h_{c,j}(t)$—domain function, which usually uses a Gaussian function, i.e.,:

$$h_{c,j}(t) = \exp\left(-d_{cj}^2 / \left(2r(t)^2\right)\right) \tag{30}$$

where $d_{cj}$—the distance between neuron *c* and any activated neuron *j*; *r*—the domain radius.

### 5.3. Quadratic Decomposition

Singular spectrum analysis (SSA) is a preprocessing method of time series data that combines multivariate statistics and probability theory. The one-dimensional time series $x = \{x_i | i = 1, 2, \ldots, N\}$ is embedded into the trajectory matrix $X$, which is:

$$X = \begin{bmatrix} x_1 & x_2 & \cdots & x_K \\ x_2 & x_3 & \cdots & x_{L+1} \\ \cdots & \cdots & \cdots & \cdots \\ x_L & x_i & \cdots & x_N \end{bmatrix} \tag{31}$$

where *N*—sequence length; *L*—window length; *K*—calculation length, i.e., the number of columns of the *X* matrix, with $K = N - L + 1$.

### 5.4. Power Prediction Algorithm Framework

The overall framework flow of the power prediction algorithm framework is shown in Figure 13. The fluctuation characteristics of PV power under different weather conditions differ. In order to obtain a more accurate power prediction, this paper uses the SOM clustering method to classify the original data into the following weather types—sunny-day data, overcast- and rainy-day data, and sudden-change-day data.

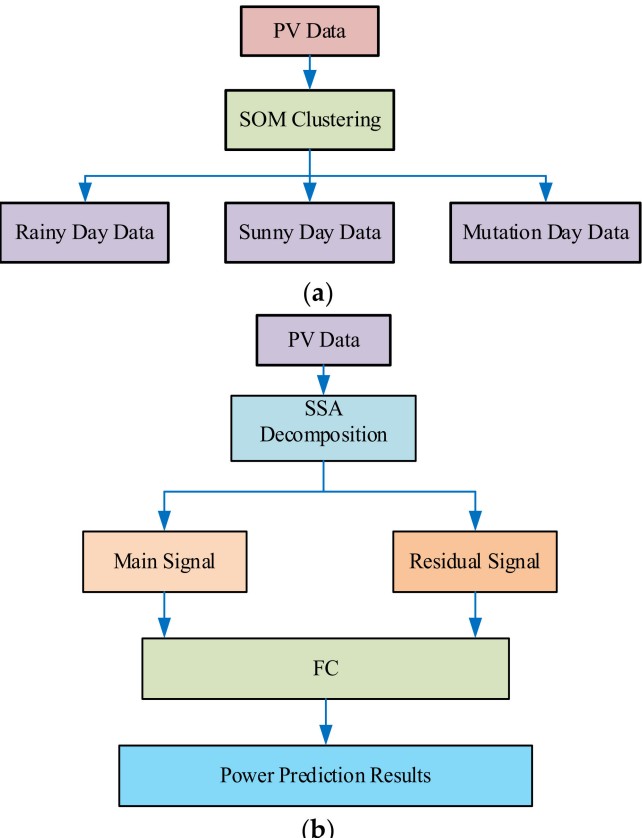

**Figure 13.** PV power prediction process. (**a**) SOM clustering process. (**b**) Secondary decomposition and power prediction.

## 6. Simulation Example Validation

Through the above theoretical elaboration and analysis, a three-machine and nine-node system model, as depicted in Figure 14, is established in PSCAD/EMTDC to verify the theory's rationality. The simulated system consists of three synchronous generating units, $G_1$, $G_2$, and $G_3$, with a capacity of 300 MW, a PV module device with a capacity of 50 MW, and three loads.

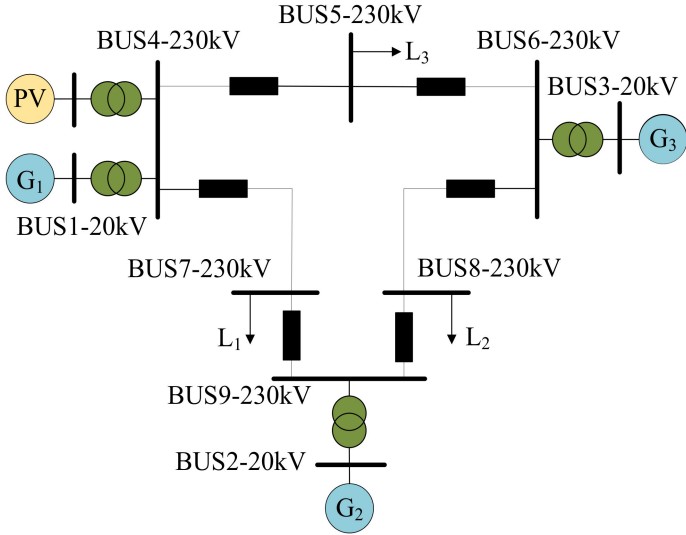

**Figure 14.** The three-machine and nine-node simulation model.

### 6.1. Analysis of the Inertial Response of a System with Different Control Parameters

According to the established three-machine and nine-node simulation model of the additional PV module equipment, the simulation analysis is carried out before and after the FR control strategy is applied. During the simulation, the light intensity is assumed to be constant at 1000 W/m$^2$, a load consuming 70 MW of active power is put in at $t$ = 4 s, the capacity of the PV module equipment is 50 MW, and the PV module is left with 20% of spare power, with the simulation comparison results displayed in Figures 15–17.

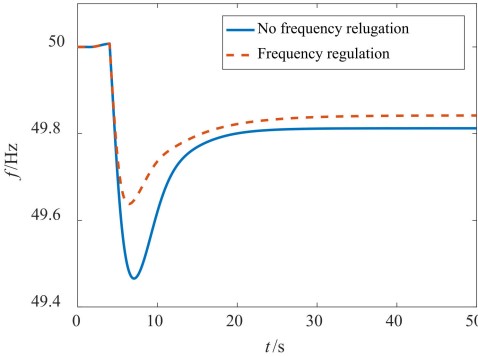

**Figure 15.** System frequency response with and without improved strategy.

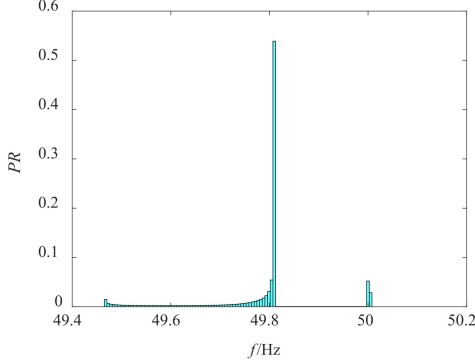

**Figure 16.** Normal distribution for system frequency response under no improved strategy.

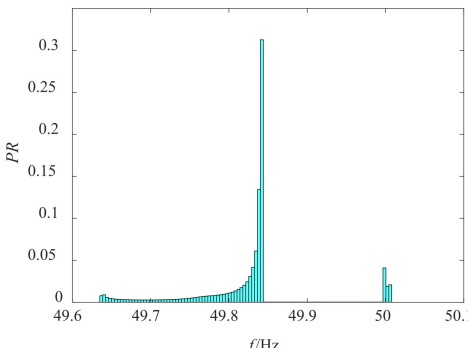

**Figure 17.** Normal distribution for system frequency response under improved strategy.

The simulation comparison results indicate that the system frequency drops when the load is put in at $t$ = 4 s. When the FR control strategy is not applied, the lowest value of frequency drop is 49.466 Hz and the final steady-state value of frequency is 49.812 Hz; after the FR control strategy is applied, the lowest value of frequency drop is 49.637 Hz and the final steady-state value of frequency is 49.841 Hz. Compared to before frequency regulation, the system reaches stability at approximately $t$ = 45 s after 41 s of dynamic adjustment, with an increase in the minimum value of frequency of approximately 0.171 Hz, a reduction in the maximum variation deviation of frequency, and an increase in the final steady-state value of frequency. The same can be seen from the normal distribution of the system frequency, when the proportion of high frequencies in the system increases significantly after the frequency control.

When a load consuming 70 MW of active power is put in at $t$ = 4 s, the simulation results for the system frequency, the PV equipment's output power, and the output power of the synchronous generator $G_2$ when the regulation factor $k$ is changed are pictured in Figure 18.

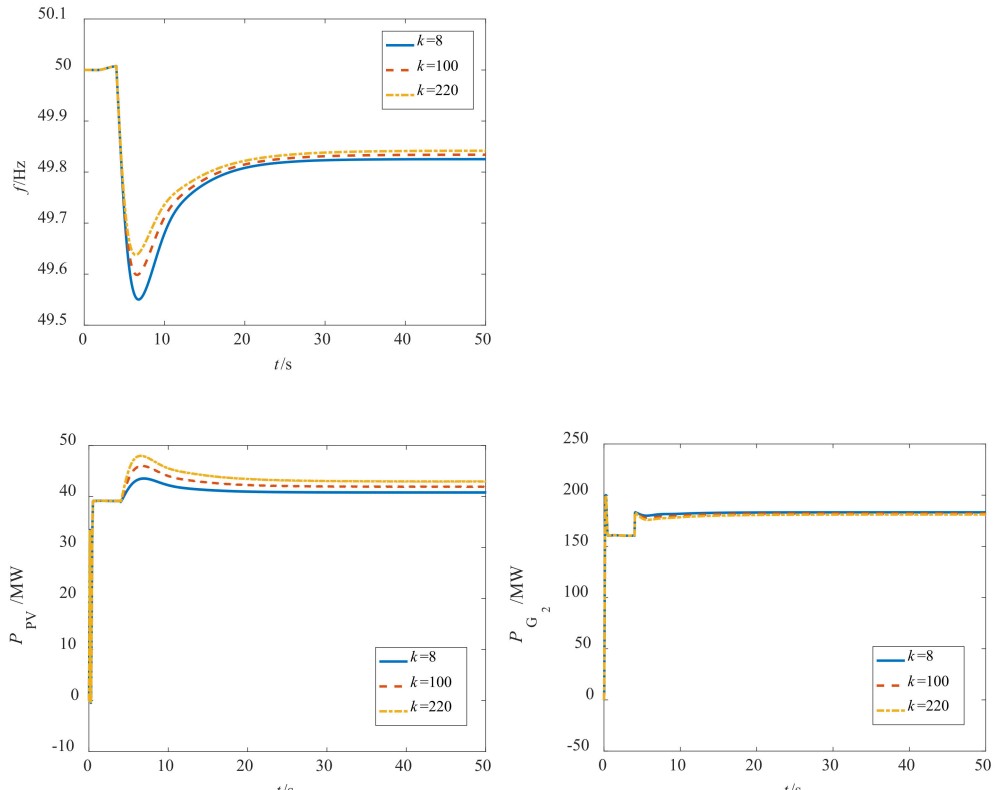

**Figure 18.** System response with different regulation coefficients.

When the regulation coefficient *k* is taken as 8, 100, and 220, the system frequency rises continuously, and the PV facility exploits the reserve power to constantly generate incremental active power to suppress the frequency drop. The lowest value of frequency drop is 49.55 Hz when *k* = 8; the lowest value of frequency drop is 49.599 Hz when *k* = 100; the lowest value of frequency drop is 49.637 Hz when *k* = 220. Therefore, as the regulation coefficient increases, the greater the inertial power provided by the PV device, and the maximum deviation in frequency variation decreases and the ultimate steady-state value of frequency increases.

When a load consuming 70 MW of active power is put in at *t* = 4 s, the simulation results for system frequency, PV device output power, and synchronous generator $G_2$ output power are displayed in Figure 19 when the virtual inertia time constant $H_{dc}$ is varied.

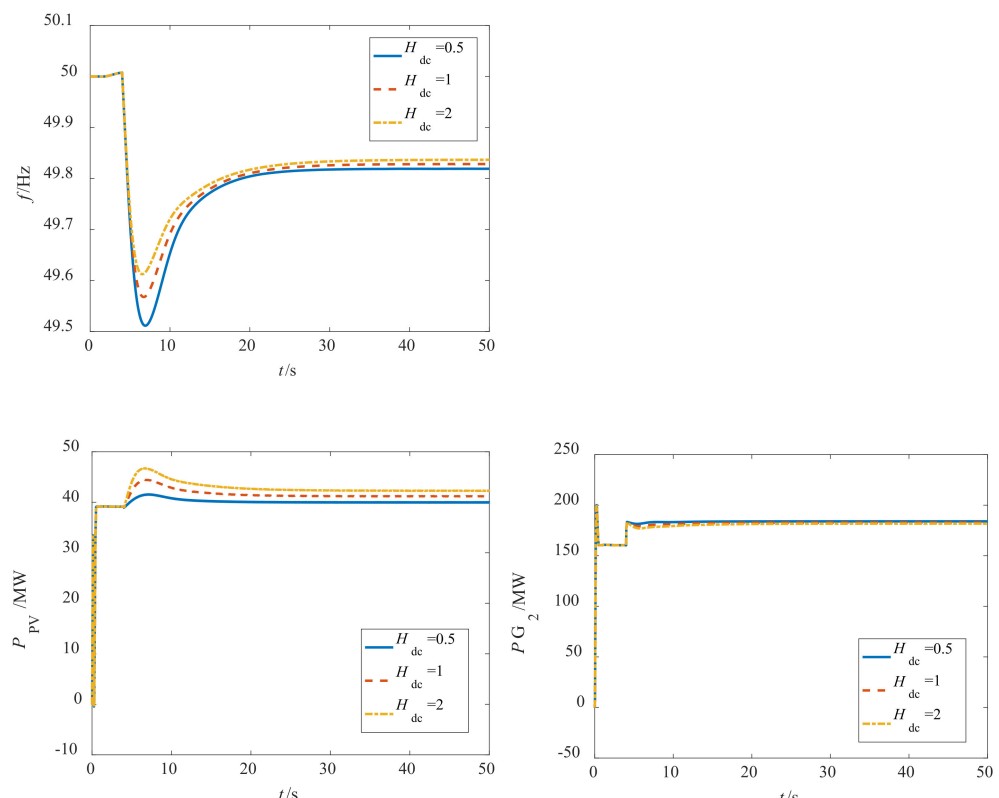

**Figure 19.** System response for different virtual inertia time constants.

As the virtual inertia time constant $H_{dc}$ increases sequentially, the lowest values of the system frequency drop are 49.511 Hz, 49.568 Hz, and 49.612 Hz, respectively, with it observed that the system frequency drop has slowed down and deviated less from the reference value. The PV installation output power can accurately respond to the system frequency variation, supplying effective inertia power to lift the system frequency, which is conducive to the steady system operation.

## 6.2. Analysis of the Inertial Response of the System during Fluctuations in Light Intensity

The previous section's environmental conditions for the simulated operation were carried out at a constant light intensity. Considering that the PV module equipment in the actual operation of the grid is affected by environmental factors such as light intensity, the PV output power will vary in size and, therefore, the system frequency will fluctuate accordingly. As depicted in Figure 20, the light intensity is set to fluctuate once every 0.5 s interval.

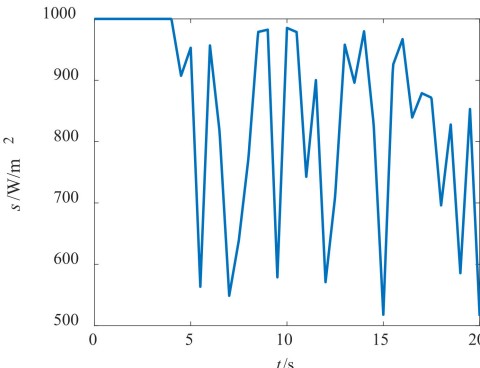

**Figure 20.** System time domain diagram for the light intensity.

The system's dynamic response at different light intensities is shown in Figure 21. When the PV equipment does not participate in system frequency regulation, the active power deficit brought about by its output power fluctuation will be fully borne by the synchronous generator set. When the PV equipment actively participates in frequency regulation, the PV equipment makes up for the power deficit before frequency regulation by increasing the output power, alleviating the system frequency drop and the overall change in system frequency drop. The deviation is reduced and the stability of the system operation is improved.

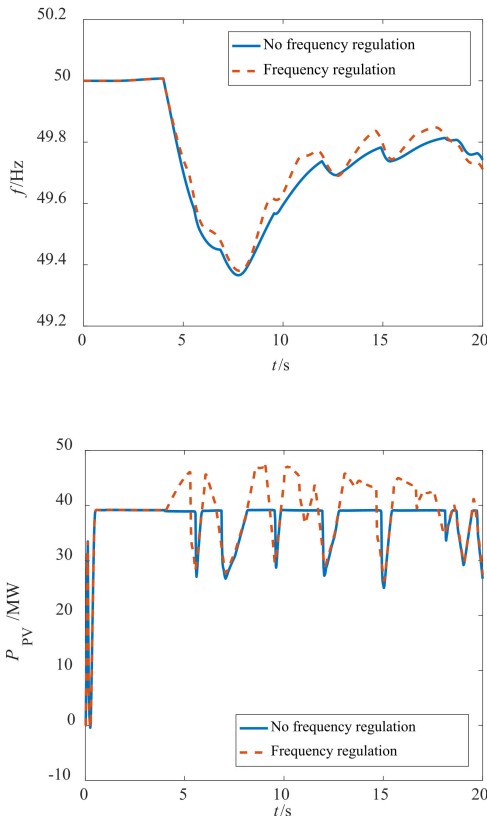

**Figure 21.** System response under fluctuations in light intensity.

When $k = 8$, observing the frequency waveform in Figure 22, it can be seen that the rate of change of frequency and the performance of the dropout nadir under the proposed control strategy has improved significantly compared to the literature [27], with the lowest frequency dropout value increasing from 49.55 Hz to 49.641 Hz and a corresponding increase in the final value of frequency at steady state, indicating that the proposed control can improve the inertia of the system while providing a primary frequency modulation level.

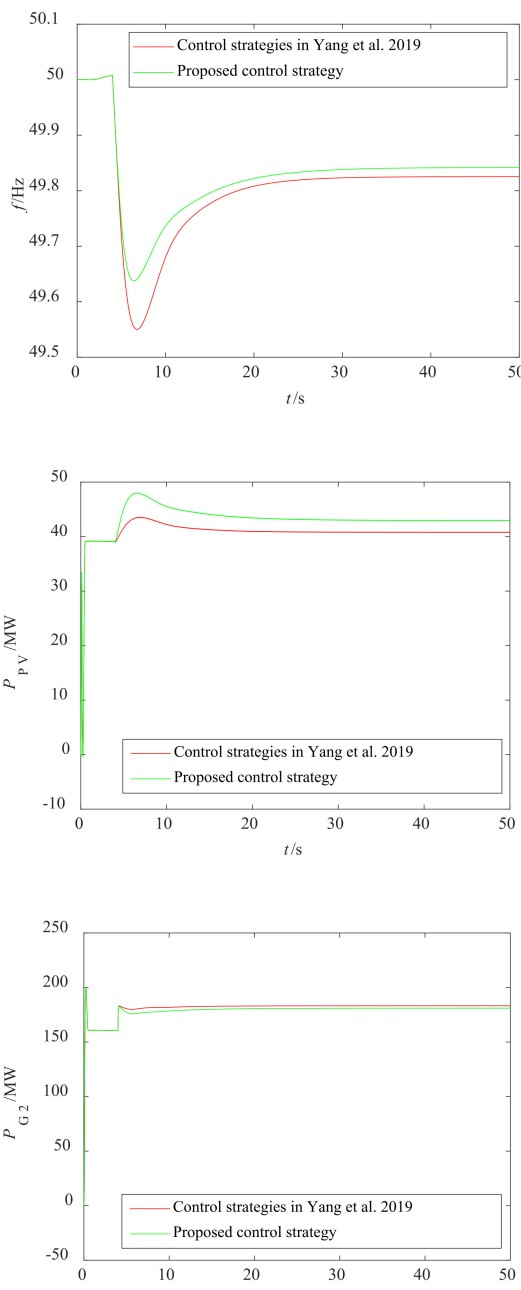

**Figure 22.** Comparison of system response before [27] and after FR control strategy improvement.

Further observation of the output power waveforms of the PV and synchronous generator $G_2$ in Figure 22 reveals that the proposed control can make the PV system respond quickly to changes in system load and release the backup energy of the PV system quickly to increase the output power. Since the inertial response is different from that of the control in the literature [27], their transient processes will also be different, and the PV output peak power under the proposed control strategy is the largest to reduce the frequency shift of the system. At steady state, the proposed control performs similarly to that of the literature [27], with the maximum steady-state power at the PV output and the minimum steady-state power at the synchronous generator output, providing a primary frequency regulation for the system and reducing the standby load shedding rate. Simulation results show that the control strategy proposed in this paper improves the frequency response performance of the system and the effect is better than that of the literature [27].

As the PV penetration rate increases, the grid-connected PV capacity is 40 MW, 45 MW, and 50 MW, with a constant load of 70 MW at $t$ = 4 s. Observing the system response curve in Figure 23, it can be seen that as the PV penetration rate increases, the system frequency minimum dip value and the final steady-state value increase, the PV output active power increases, and the synchronous generator output active power decreases. As PV penetration increases, the frequency characteristics of the power system will depend more on the FR capacity and FR capability that PV can provide. PV should participate in FR and provide sufficient FR capacity in order to curb the decline in system frequency characteristics, so PV participation in system FR is crucial to curb the decline in system frequency characteristics.

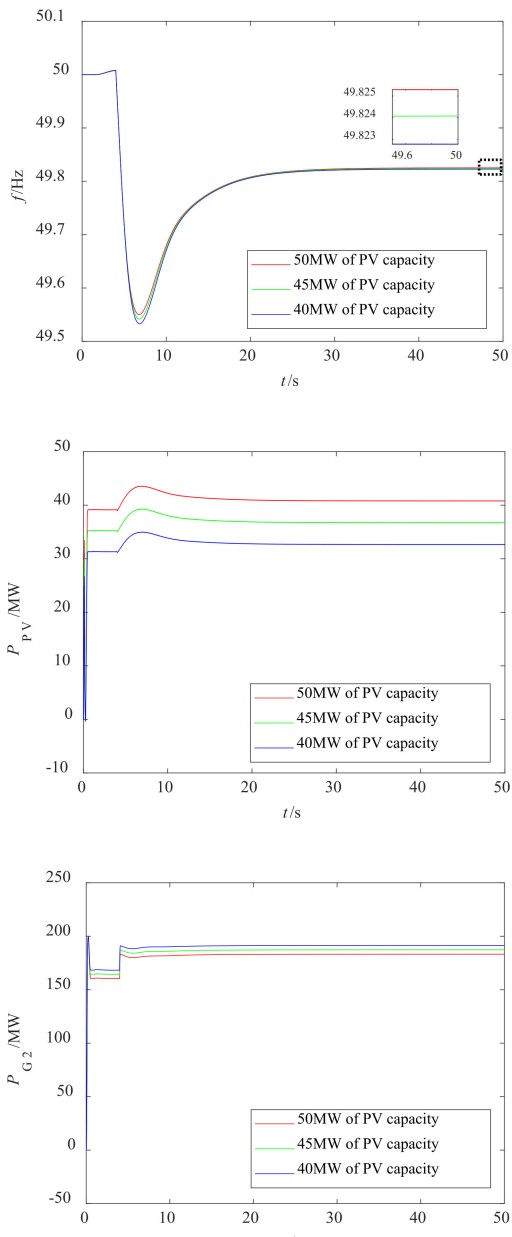

**Figure 23.** Comparison of system response at different PV penetration rates.

## 7. Conclusions

In this paper, an effective control strategy for grid-connected inverters based on PV access to the distribution network is proposed in order to solve the problems of stability and insufficient FR resources during the regular operation of two-stage PV power systems. The

dynamic process of the system frequency response is analysed, the operating mechanism of the system in the left and right halves of the PV operating curve is revealed through a grid frequency response model, and the influence of parameters such as virtual inertia time constants and damping coefficients on the attraction domain of the system is further analysed using the Lyapunov energy function method.

This control method can effectively increase the inertia of the system, allowing the PV to accurately predict the output active power and provide frequency support under different weather types and irradiance conditions, reducing power losses and frequency deviations during system disturbances, enhancing the flexibility and robustness of the distribution network. In addition, this method is not complex and allows virtual inertia control, equivalent to the frequency regulation effect of synchronous generators, to be achieved without increasing the cost of energy storage.

**Author Contributions:** Conceptualization, S.W. and H.Z.; methodology, S.W. and H.Z.; software, S.Z. and H.Z.; validation, S.W. and H.Z.; formal analysis, S.W. and H.Z.; investigation, S.W. and S.Z.; resources, S.W.; data curation, S.W. and S.Z.; writing—original draft preparation, S.W. and H.Z.; writing—review and editing, S.Z. and H.Z.; writing—language refinement, S.W. and H.Z.; visualization, S.W.; supervision, S.W.; project administration, S.W. All authors have read and agreed to the published version of the manuscript.

**Funding:** This research was funded by the Jiangsu Province industry-university-research cooperation project, BY2022056. Postgraduate Research and Practice Innovation Program of Jiangsu Province, SJCX21_0952. Research Fund project of Nanjing Institute of Technology, CKJB202114, CXY202015.

**Institutional Review Board Statement:** Not applicable.

**Informed Consent Statement:** Not applicable.

**Data Availability Statement:** Not applicable.

**Conflicts of Interest:** The authors declare no conflict of interest.

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
