# Peer review of "Two-Stage Grid-Connected Frequency Regulation Control Strategy Based on Photovoltaic Power Prediction"

_sustainability, doi:10.3390/su15118929_

Round 1

Reviewer 1 Report

This paper proposes a two-stage grid-connected frequency regulation control strategy based on photovoltaic power prediction

1- Please check the paper to avoid any language problems and inconsistency in the font such as keywords.

2- The references in the literature review are outdated. As well, please consider more references and investigate them one by one.

3- Please use appropriate references for mathematical models.

4- Add important numerical results to the conclusion.

5- I Think this is better to show the operation of the proposed method under high penetration of solar power. you can add a new section for that and increase the pv portion step by step (power of pv/load demand).

6- A sensitivity analysis should be considered for optimal settings of the controller. How did you select the input data?

1- Please check the paper to avoid any language problems and inconsistency in the font such as keywords.

Author Response

Dear editors and dear reviewers:

Thank you for your letter and the reviewers’ comments concerning our manuscript. Those comments are valuable and very helpful. We have read through comments carefully and have made corrections. Based on the instructions provided in your letter, we uploaded the file of the revised manuscript. The amendments in the manuscript have been marked as follows.

We would love to thank you for allowing us to resubmit a revised copy of the manuscript and we highly appreciate your time and consideration.

Sincerely.

Reviewer 1:

Q1. Please check the paper to avoid any language problems and inconsistency in the font such as keywords. 

Response:

The situation you mentioned has been modified and marked in the paper.

Q2. The references in the literature review are outdated. As well, please consider more references and investigate them one by one. 

Response:

The paper has considered more references in recent years and has been updated.

Q3. Please use appropriate references for mathematical models.

Response:

The equations of the PV array output current IPV and output power PPV versus output voltage VPV are as follows[24]:

                     (1)

Where Isc and UOC are short-circuit current and open-circuit voltage for the PV panel, and C1 and C2 are coefficients to be determined.

As shown in Figure 6, M and N are the power points on the left and right sides of the PV array operation when the PV system leaves a certain amount of backup power. When there is an imbalance between the power output of the synchronous generator set and the load power, the system frequency will fluctuate significantly. Under the action of the unbalanced power, the equation of rotor motion in the synchronous generator set is illustrated in equation (7)[26]:

                       (7)

Where: δ is the rotor work angle, ω0 is the rated electrical rotor angular velocity, ω is the rotor electrical angular velocity and HSG is the rotor inertia time constant.

  1. Q Jia, G Yan, Y Lin, et al. Stability analysis of multiple paralleled photovoltaic power generation units connected to weak AC system. Automation of electric power systems. 2018, 42(3): 14-20.
  2. X Qin, L Su, Y Chi, et al. Functional orientation discrimination of inertia support and primary frequency regulation of virtual synchronous generation in large power grid. Automation of electric power systems. 2018, 42(9): 36-43.

Q4. Add important numerical results to the conclusion.

Response:

In this paper, the system frequency response characteristics of the proposed control strategy are mainly qualitative analysis, and there is not much quantitative analysis.

Q5. Think this is better to show the operation of the proposed method under high penetration of solar power. You can add a new section for that and increase the pv portion step by step (power of pv/load demand).

Response:

Figure 23. Comparison of system response at different PV penetration rates

As the PV penetration rate increases, the grid-connected PV capacity is 40MW, 45MW and 50MW, with a constant load of 70MW at t=4s. Observing the system response curve in Figure 23, it can be seen that as the PV penetration rate increases, the system frequency minimum dip value and the final steady state value increase, the PV output active power increases and the synchronous generator output active power decreases. As PV penetration increases, the frequency characteristics of the power system will depend more on the FR capacity and FR capability that PV can provide. PV should participate in FR and provide sufficient FR capacity in order to curb the decline in system frequency characteristics, so PV participation in system FR is crucial to curb the decline in system frequency characteristics.

Q6. A sensitivity analysis should be considered for optimal settings of the controller. How did you select the input data?

Response:

From Figure 8, the transfer function G1(s) from the load perturbation ΔPL to the frequency perturbation Δfg is as follows:

                (15)

Therefore, compared to the conventional grid frequency response model, the essence of this control strategy of frequency regulation is to change the damping factor D of the synchronous generator set, the new damping factor becomes D+akfk.

According to the equation (15), under different control coefficients k, the relationship between the initial value df/dt of grid frequency change rate and the minimum drop value fpeak of grid frequency and the inertial time constant Hdc is shown in Figure 9. From Figure 9, it can be seen that the change of control coefficient k has no effect on the initial value of grid frequency change rate; as the time constant Hdc increases, the lowest frequency drop increases.

Figure 9. Relation curve between frequency characteristic quantity and Hdc

Reviewer 2 Report

Well-written work on frequency regulation control strategy based on the dynamic characteristics of the grid-side DC capacitor, to improve the stability of the distribution network. But the following points if addressed can make this paper fit for publication:

1. "Carrying capacity" is suggested to be improved in the abstract but there is no mention of the 'carrying' capacity in the discussion or the conclusion.

2. What is the 'SOM' clustering algorithm and why is it used here? There is no justification provided. In addition, what is the full form of SOM?

2. Literature review section is missing. Authors are recommended to review the following as well: 10.1109/ACCESS.2021.3132223, 10.1109/TIE.2019.2899546.

Reviewer 3 Report

Dear Authors.

The paper is of high interest to the scientific community and should be generated in two parts - if the publisher allows it. A first paper to lay the foundations of modeling and control strategies in photovoltaic systems and their influence on the frequency of the electrical network - review paper. In a second paper in which the control strategy carried out is described in detail in the sense that each parameter that is introduced to the model must be reasoned based on the previous work of the antecedents that is improved.

If finally two parts are not generated and in the meantime you must do the following:

1) Explain better the origin of equations (2) to (4) and why precisely the parameter "r" is introduced in that place in the control system.

2) Regarding the conventional control for PV inverter: they must decide if figure 6 is essential or not.

3) Section 4 is the most important because it describes the control strategy used. It should indicate the most important parameters of the models used, emphasizing the improvements introduced and the previous models and strategies that are improved.

4) In the above sense, make a better review of previous works in which the improvements introduced in this work are revealed. That is, cite a reference and immediately, then say what would be improved.

5) What parameter of the model indicates the rate of penetration of the resource and what is the background related to it?

Receive a warm greeting.

The paper is written in an intelligible English, which in may opinion is enough. In a subsequent stage, language may be polished or improved. 

Round 2

Reviewer 1 Report

I am satisfied

I am satisfied

Reviewer 3 Report

Authors have addresssed queries an improved the paper.

fine